# Vibrational stress affects extracellular signal-regulated kinases activation and cytoskeleton structure in human keratinocytes

Dongjoo Kim 1,2, Soonjo Kwon 1*

1 Department of Biological Engineering, Inha University, Incheon, Korea, 2 Biology and Medical Device Evaluation Team, Korea Testing & Research Institute, Gwacheon, Korea

* soonjo.kwon@inha.ac.kr

**Data Availability Statement:** All relevant data are within the manuscript and its Supporting Information files.

**Funding:** This study was supported by the National Research Foundation of Korea (NRF-

## Abstract

As the outermost organ, the skin can be damaged following injuries such as wounds and bacterial or viral infections, and such damage should be rapidly restored to defend the body against physical, chemical, and microbial assaults. However, the wound healing process can be delayed or prolonged by health conditions, including diabetes mellitus, venous stasis disease, ischemia, and even stress. In this study, we developed a vibrational cell culture model and investigated the effects of mechanical vibrations on human keratinocytes. The HaCaT cells were exposed to vibrations at a frequency of 45 Hz with accelerations of 0.8g for 2 h per day. The applied mechanical vibration did not affect cell viability or cell proliferation. Cell migratory activity did increase following exposure to vibration, but the change was not statistically significant. The results of immunostaining (F-actin), western blot (ERK1/2), and RT-qPCR (FGF-2, PDGF-B, HB-EGF, TGF-β1, EGFR, and KGFR) analyses demonstrated that the applied vibration resulted in rearrangement of the cytoskeleton, leading to activation of ERK1/2, one of the MAPK signaling pathways, and upregulation of the gene expression levels of HB-EGF and EGFR. The results suggest that mechanical vibration may have wound healing potential and could be used as a mechanical energy-based treatment for enhancing wound healing efficiency.

## Introduction

The skin is a complex, multilayered organ and is composed of epidermis at the outer surface and dermis, the connective tissue under the epidermis [1,2]. The epidermis is mainly comprised of keratinocytes, while the dermis is made up of fibroblasts and the extracellular matrix (ECM). As the outermost organ, the skin provides a wall-like barrier function to defend against physical, chemical, and microbial assaults and to prevent water release from the inside to the outside of the body [3]. The skin can be damaged by injuries such as wounds or bacterial and viral infections, and a damaged skin barrier should be adequately and rapidly restored to protect the body from physical, chemical, and biological hazards. Wound healing is a complex process that can be divided into three overlapping but distinct phases; 1) inflammation, 2)

2017R1D1A1B03029589) and an Inha University
Research Grant, Korea (61529-01) to SK.

**Competing interests:** The authors have declared
that no competing interests exist.

proliferation, and 3) tissue remodeling [4,5]. The wound healing process can be delayed or prolonged by health conditions such as diabetes mellitus, venous stasis disease, ischemia, and even stress [6]. In particular, diabetic or venous ulcers in lower limbs are major sources of chronic wounds, leading to poor quality of life, high health care cost, and a long-term treatment period [7,8].

There are myriad methods for treating chronic wounds, including debridement, pressure offloading, dressings, and topical uses of antibiotics and growth factors [9,10]. These standard treatments are often combined with mechanical energy-based therapies such as electrical stimulation, negative pressure, ultrasound, and vibration [11–16]. Although there is evidence showing that treatments using mechanical stimulation improve blood circulation, angiogenesis, and wound healing [17], there is debate on several issues related to the quality of the evidence and to the actual efficacy of such treatments [18,19]. Further, investigative approaches to elucidate the cellular mechanisms of mechanical energy-based therapies have not been fully described.

We hypothesized that the wound healing effects of mechanical vibrations may be related to the activation of mechano-transduction signaling pathways. Previous studies have suggested that low-magnitude high-frequency vibration (LMHFV) can activate mitogen-activated protein kinase (MAPK) pathways [20,21], which are essential for regulation of cytophysiological processes such as cell growth, cell proliferation, cell differentiation, metabolic activity, inflammatory response, and apoptosis [22–24]. In the current study, we designed a vibrational cell culture model, enabling us to test our hypothesis and to investigate the cellular mechanisms involved in the wound healing effects of mechanical stimulation. A 45 Hz vibration at 0.8 $g$ was applied to human adult low-calcium high-temperature keratinocytes (HaCaT), and the vibrational effects on cell viability, cell proliferation, cell migration, cytoskeletal structure, activation of extracellular signal-regulated kinases (ERKs), and gene expressions of growth factors and growth factor receptors were explored. We observed that the applied mechanical vibration did not significantly affect cell proliferation or migration, but it did alter cytoskeletal structures, activation of ERKs, and upregulation of gene expressions of some growth factors and their receptors.

## Materials and methods

### Vibrational culture model design

The vibrational culture model was designed and fabricated to study the effects of mechanical vibration on cellular responses (Fig 1). It consists of an acrylic frame (15 cm × 25.4 cm × 8.5 cm, width × length × height), a sample holder, a voice coil actuator (VCA; BEI Kimco, Vista, CA, USA) and a digital servo driver (Pluto, Ingenia, Motion Control, Barcelona, Spain). The model is powered by a direct current (DC) supply and controlled by software (MotionLab, Ingenia) running on a laptop computer. The software allows the user to control and monitor the frequency and displacement of the applied vibration. When placed on the sample holder, a standard multi-well cell culture plate is oscillated up and down along the vertical axis by the VCA. The acrylic frame, VCA, and sample holder were installed inside a $CO_2$ incubator, while the digital servo driver, DC power supply, and the laptop computer were located outside the $CO_2$ incubator.

### Cell culture

The HaCaT cells were kindly provided by CHA University (Seongnam, Korea). They were routinely cultured at 37˚C with 5% $CO_2$ in T75 flasks. Dulbecco's modified Eagle's medium (Gibco, Carlsbad, CA, USA) supplemented with 10% fetal bovine serum (Gibco) and 1% penicillin-streptomycin (Gibco) was used as the culture medium. For vibration exposure, HaCaT cells (passage 43) were trypsinized from a T75 flask and seeded onto 6-well culture plates or

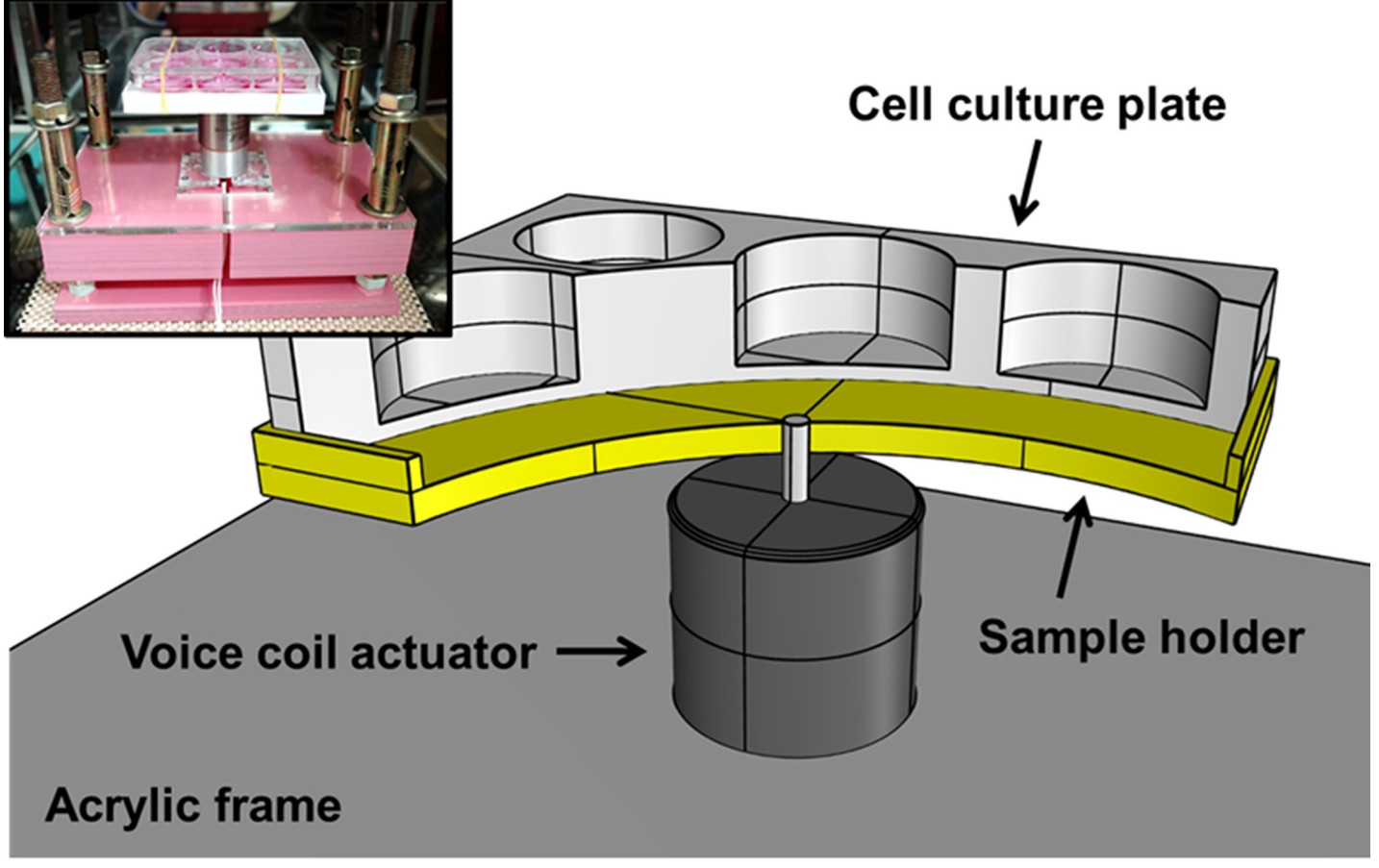

**Fig 1. Schematic design and photographic image of the vibrational culture model.** The culture model is composed of an acrylic frame, a voice coil actuator (VCA), a sample holder, and a digital servo driver. The acrylic frame with the VCA and the sample holder are located inside the $CO_2$ incubator, while the digital servo driver connected to the DC power supply and the laptop computer are located outside the incubator.

4.55 cm$^2$ cell culture slides (SPL Life Science, Gyeonggi-do, South Korea) at a density of 50,000 cells/cm$^2$, and culture media were replenished every 2 days. For the cell proliferation analysis, cells were seeded onto 12-well culture plates at a density of 3,000 cells/cm$^2$.

## Vibration exposure

The above described vibrational culture model was used to apply a 45 Hz mechanical vibration at 0.8 *g* to HaCaT cells. For the cell viability and immunofluorescence staining analyses, the vibration was applied once, and for 2 h only, after HaCaT cells reached confluence (approximately 2 days after seeding). For the cell proliferation assay, HaCaT cells were exposed to vibration for 2 h per day for 5 days beginning 1 day after seeding. For the other analyses, HaCaT cells were cultured until reaching confluence, after which mechanical vibration was applied for 2 h per day for 5 days. The control group was cultured separately under the same conditions but without vibration exposure.

## Cell viability

The trypan blue dye exclusion assay was performed to investigate the effects of mechanical vibration on cell viability. Following exposure to vibratory stimulation, the HaCaT cells were

washed with Dulbecco's phosphate-buffered saline (DPBS; Gibco), detached by trypsin-ethyl-enediaminetetraacetic acid (Trypsin-EDTA; Gibco), and mixed with 0.4% trypan blue staining solution (Gibco). The stained cells were counted by using a hemocytometer and an inverted microscope (CKX 53, Olympus, Tokyo, Japan). Cell viability was calculated by dividing the number of viable cells by that of total cells.

## Cell proliferation

Cell counting kit-8 (CCK-8; Dojindo Laboratories, Kumamoto, Japan) was used to measure the effects of vibration on cell proliferation. Before applying the vibratory stimulation, 30 μL of CCK-8 solution was added to the media and the HaCaT cells incubated for 2 h. Next, 0.15 mL of the culture media was transferred to a 96-well plate, and the absorbance was measured at 450 nm using a microplate reader (Multiskan GO, Thermo Fisher Scientific, Waltham, MA, USA). The HaCaT cells were then washed twice with DPBS and exposed to mechanical vibration for 2 h. The above procedures were repeated for 6 days until the last absorbance measurement was acquired.

## *In vitro* scratch assay

*In vitro* scratch assays were carried out to determine the vibrational effects on cell migration. The HaCaT cell monolayers were scratched by a sterile 1 mL pipette tip and washed twice with DPBS to remove cell debris. Next, the scratched areas were microphotographed by using an Olympus CKX 53 microscope with a charge-coupled device camera (DP80, Olympus), and the HaCaT cells were exposed to mechanical vibration for 2 h per day. The obtained images were processed using cellSens software (Olympus). Analysis of cell migration was performed by measuring the lengths of the vertical straight lines connecting the borders of the cell-free regions. The closed width was calculated by subtracting the average length of the vertical straight lines measured at 96 h after scratching from that measured immediately after scratching.

## Immunofluorescence staining

Fluorescence microscopy was performed to observe structural changes in the cytoskeleton of HaCaT cells. After the HaCaT cells underwent vibration exposure, culture media were removed and the cells rinsed with DPBS and fixed with 4% paraformaldehyde (Sigma, St Louis, MO, USA). The fixed cells were permeabilized with 0.1% Triton X-100 in DPBS and washed thrice with DPBS. The HaCaT cells were then incubated with AlexaFluor 488 conjugated phalloidin (ab176753, Abcam, Cambridge, UK) for 90 min. The stained cells were rinsed thrice with DPBS and mounted with mounting solution (Vectashield H-1200, Vector Laboratories, Burlingame, CA, USA) containing 4′,6-diamidino-2-phenylindole (DAPI) for counterstaining. Fluorescence images were obtained by using an Olympus CKX 53 microscope with a DP80 camera and were processed by using cellSens software.

## Immunoblotting analysis

Western blot analysis was carried out to assess the activation of ERKs following exposure to mechanical vibration. After completion of the vibration protocols, culture media were aspirated and the HaCaT cells were washed twice with ice-cold DPBS before being scraped in radio-immunoprecipitation assay buffer (EBA-1149, Elpis Biotech, Daejeon, Korea) supplemented with protease and phosphatase inhibitor cocktail (78441, Thermo Fisher Scientific). Cell extracts were incubated at 4˚C for 30 min with gentle agitation. They were then

centrifuged at 14,000 *g* for 10 min, and the supernatants were collected. Protein concentration was determined by using a bicinchoninic acid protein assay kit (23227, Thermo Fisher Scientific) according to the manufacturer's manual. The supernatants were diluted in Laemmli sample buffer and boiled at 70°C for 10 min. Each 30 μg sample of total protein was separated by electrophoresis through 10% sodium dodecyl sulfate -polyacrylamide gel and then transferred to a polyvinylidene difluoride membrane (162–0177, Bio-Rad, Hercules, CA, USA) using a semidry transfer apparatus (Bio-Rad). Nonspecific binding was blocked by using 5% skim milk in Tris-buffered saline for 1 h at room temperature, and the membrane was then incubated with primary antibodies against total ERK 1/2 (9102, Cell Signaling Technology, Danvers, MA, USA), phospho-ERK 1/2 (9101, Cell Signaling Technology), or glyceraldehyde-3-phosphate dehydrogenase (GAPDH; ab9485, Abcam) overnight at 4°C. After the membrane was immersed in horseradish peroxidase conjugated secondary antibody (ab6721, Abcam) for 1 h at room temperature, blots were visualized by applying enhanced chemiluminescence reagents (34577, Thermo Fisher Scientific) and quantified using a chemiluminescence imaging system (G:BOX Chemi XRQ, Syngene, Cambridge, UK).

## Gene expression analysis

A real-time quantitative polymerase chain reaction (RT-qPCR) method was used to analyze the changes in gene expressions of growth factors and their receptors with or without vibrational stimulation. Following completion of the vibration protocols for 5 days, total RNA was extracted by using TRIzol reagent (Life Technologies, Carlsbad, CA, USA) according to manufacturer's guide. The concentration of the extracted total RNA was measured using a microspectrophotometer (DS-11, DeNovix, Wilmington, DE, USA) and each 1 μg RNA sample was reversely transcribed into cDNA by using a PrimeScript RT reagent kit (PR037A, Takara, Shiga, Japan). The real-time PCR system (CFX-96, Bio-Rad) was operated with TB Green Premix Ex Taq II (PR820A, Takara). The target genes were fibroblast growth factor 2 (FGF-2), platelet-derived growth factor subunit B (PDGF-B), heparin-binding epidermal growth factor-like growth factor (HB-EGF), transforming growth factor beta 1 (TGF-β1), epidermal growth factor receptor (EGFR), and keratinocyte growth factor receptor (KGFR; also known as fibroblast growth factor receptor 2 IIIb). GAPDH was used as the control gene. Primer sequences for each target gene are listed in Table 1. Fold changes of gene expression levels were determined by using the $2^{(-\Delta\Delta CT)}$ method [25], and melting curve analysis was carried out to assess the formation of undesired PCR products.

## Statistical analysis

All presented measurements were independently repeated at least three times, and the results are displayed as mean ± standard error of the mean values. Student's *t*-test (two-tailed) was used to determine the statistical significance of differences between the vibration-exposed group and the control group. $P < 0.05$ was considered statistically significant.

## Results

### Characterization of vibration

The movements resulting from the vibration device were monitored by using a MotionLab software that acquires real-time VCA motion data through the digital servo driver. The applied vibration oscillated along the vertical axis at a frequency of 45 Hz, and the average peak to peak displacement was 194.2 μm (Fig 2). The acceleration of the applied vibration was

Table 1. Primer sequences of target genes.

| Target gene | Size (bp[a]) | Sequences | Tm[b] |
|:---:|:---:|:---|:---:|
| GAPDH | 120 | F: GAAATCCCATCACCATCTTCCAGG | 61.23 |
| | | R: GAGCCCCAGCCTTCTCCATG | 62.62 |
| FGF-2 | 120 | F: CCACCTATAATTGGTCAAAGTGGT | 58.74 |
| | | R: CATCAGTTACCAGCTCCCCC | 59.82 |
| PDGF-B | 126 | F: ACTGATGGGGTCGCTCTTTG | 60.04 |
| | | R: CAGGGATCAGGCAGGCTATG | 59.96 |
| HB-EGF | 298 | F: CCACACCAAACAAGGAGGAG | 58.39 |
| | | R: ATGAGAAGCCCCACGATGAC | 59.82 |
| TGF- β1 | 98 | F: CGTGGAGGGGAAATTGAGGG | 60.39 |
| | | R: CCGGTAGTGAACCCGTTGATG | 61.01 |
| EGFR | 197 | F: TGTGCCCACTACATTGACGG | 60.32 |
| | | R: GCGATGGACGGGATCTTAGG | 60.04 |
| KGFR | 182 | F: GAACCCAATGCCAACCATGC | 60.39 |
| | | R: ACGTGTGATTGATGGACCCG | 60.39 |

a Base pair

b Melting temperature (˚C)

calculated as 0.8 *g* in accordance with the mathematical relationship between the frequency, peak to peak displacement, and acceleration of a sinusoidal motion.

## Cell viability

The trypan blue dye exclusion assay was used to assess the effects of mechanical vibration on cell viability (Fig 3). The applied vibration did not cause significant cell death in HaCaT cells. Following exposure to vibration, the cell viabilities of the control and vibration-exposed groups were 98.4% and 98.9%, respectively.

## Cell proliferation

The CCK-8 assay was used to examine the effects of mechanical vibration on cell proliferation. As shown in Fig 4, cell proliferative activity was unaffected by the applied vibration. The differences in absorbance at 450 nm between the control and the vibration-exposed groups were less than 5% for all assays, and there was no statistical significance of the differences.

## *In vitro* scratch assay

An *in vitro* scratch assay was performed to estimate the vibrational effects on cell migration (Fig 5). The applied vibration induced a slight increase in cell migratory activity, but the change was not statistically significant. The closed width of the control group scratch was 858 ± 35 μm, and that of the vibration-exposed group was 924 ± 62 μm.

## Immunofluorescence

Immunofluorescence analysis was performed to investigate the vibrational effects on the cytoskeletal structures in HaCaT cells (Fig 6). The control cells showed well-organized and well-distributed cytoskeletal structures, while the vibration-exposed cells exhibited a greater amount of condensed actin bundles, and the actin fibers tended to accumulate at the peripheral side near the plasma membrane.

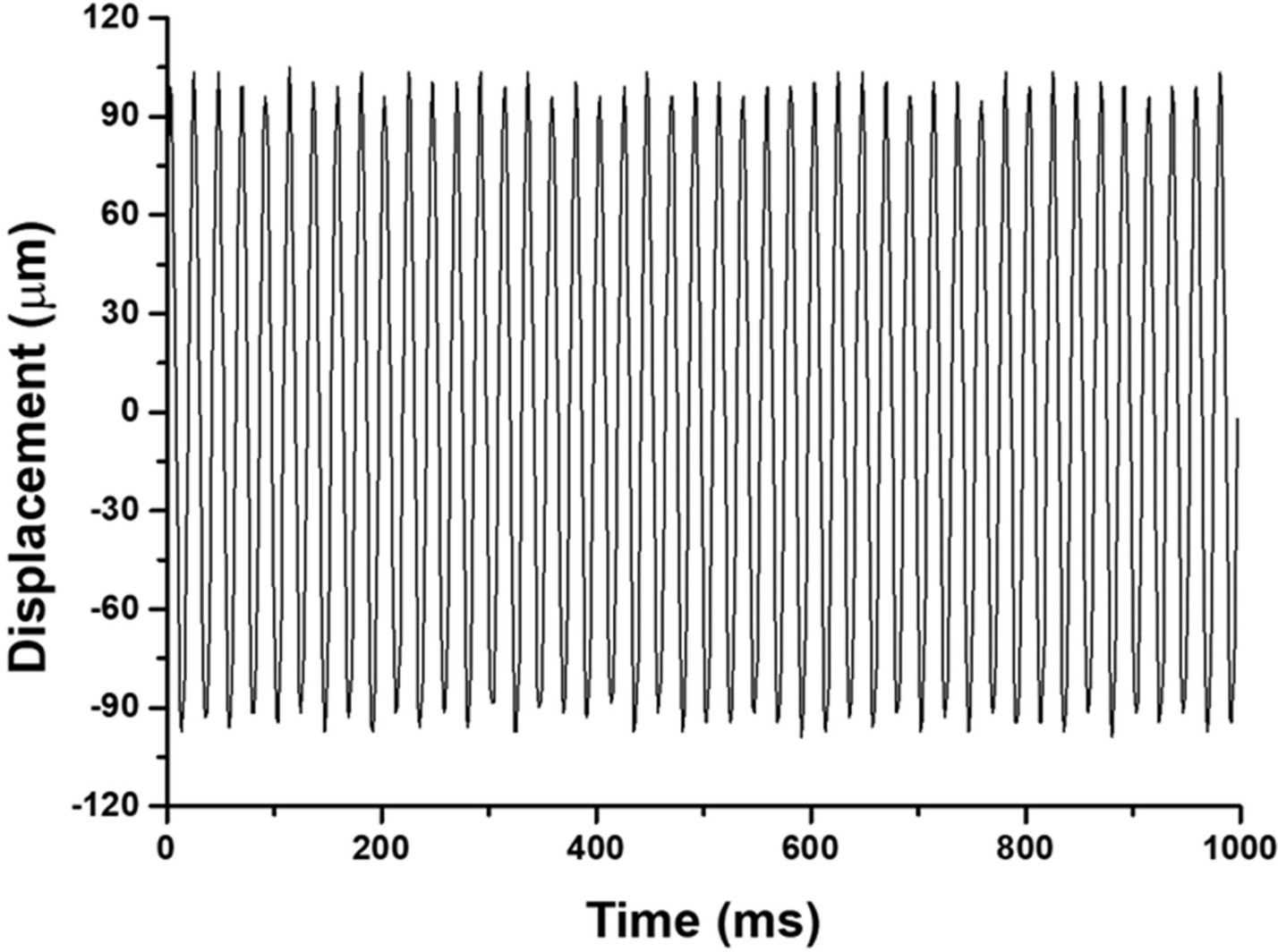

**Fig 2. Vibrational characteristics at a frequency of 45 Hz with an acceleration of 0.8 *g*.** Displacement of the sample over time was monitored by using computer software.

## Western blot

Western blot analysis was carried out to examine the effects of mechanical vibration on ERK1/2 activation (Fig 7). The results showed that the applied mechanical vibration induced a marked increase in the amount of activation of ERK1/2 in HaCaT cells. The level of phosphorylated ERK1/2 in the vibration-exposed group was 3.75 times higher than that in the control group ($p < 0.001$).

## Gene expression analysis

Gene expression levels of FGF-2, PDGF-B, HB-EGF, TGF-β1, EGFR, and KGFR were determined by performing RT-qPCR (Fig 8). Among the growth factors, the expression level of HB-EGF was 1.86-fold higher in the vibration-exposed group than in the control group ($p < 0.001$), whereas PDGF-B expression decreased by 53.5% ($p < 0.001$) following exposure to vibration. The expression of FGF-2 also decreased by 16.2%, but the change was not

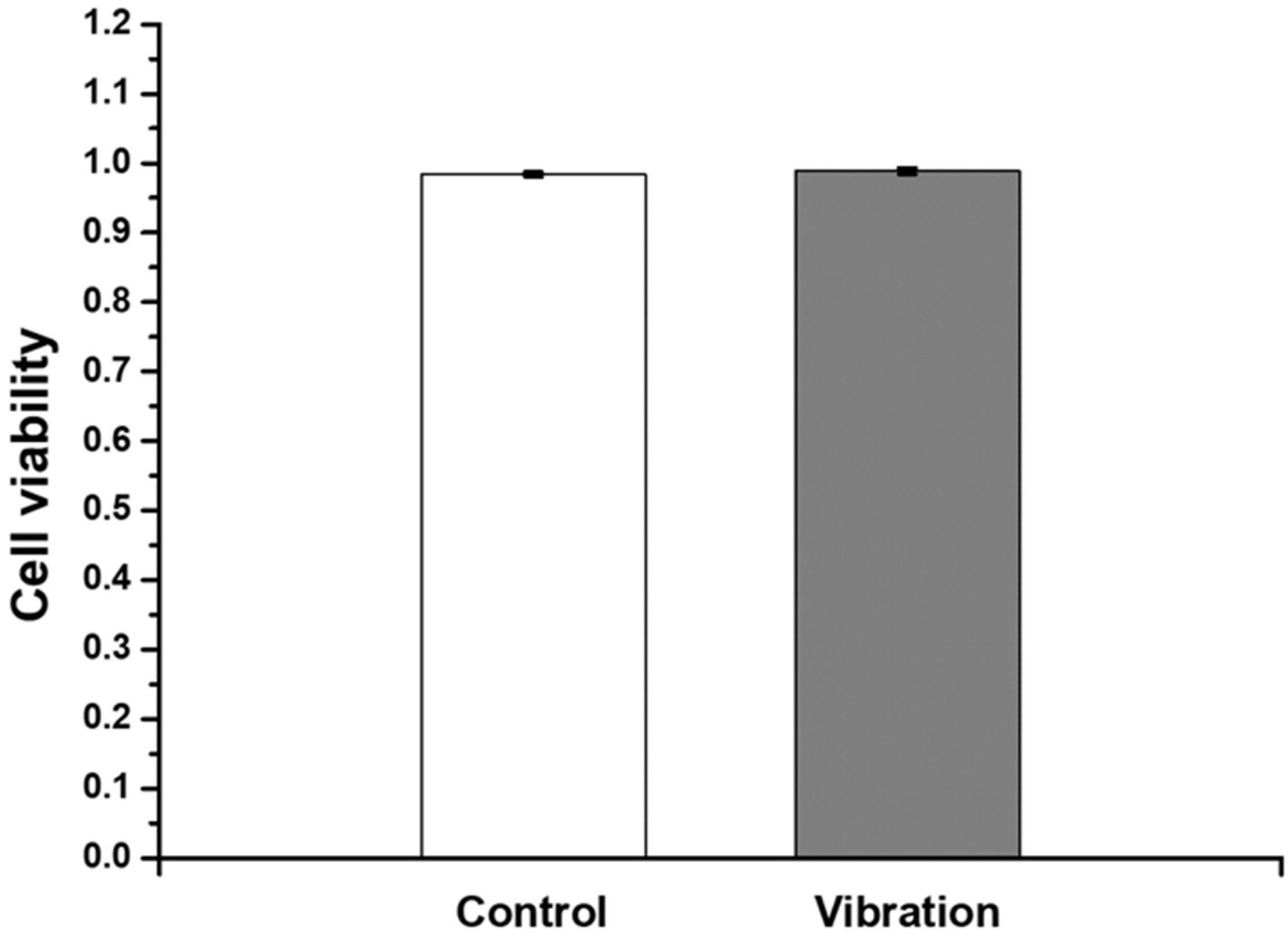

**Fig 3. Cell viability of HaCaT cells following exposure to mechanical vibration.** The viable and nonviable cells were counted, and the viability was calculated by dividing the number of viable cells by that of total cells ($n$ = 3).

statistically significant. There was no remarkable change in TGF-β1 expression. With regard to growth factor receptors, the EGFR expression level increased by 33.4% following vibration exposure ($p$ = 0.001), but the expression level of KGFR was only 1.05-fold higher in the vibration-exposed group, and the increase was not statistically significant.

## Discussion

Chronic wounds, including but not limited to diabetic foot ulcers, pressure ulcers, and venous leg ulcers have become one of the biggest healthcare challenges around the world [10,26]. Every year in the United States (US), as many as 6.5 million people suffer from these chronic wounds [27], and it has been estimated that the aggregate medical cost associated with chronic wounds exceeds US$ 25 billion per year in the US [28]. Further, chronic wounds are becoming more prevalent as the size of the elderly population, susceptible to disease such as diabetes mellitus, venous stasis disease, and ischemia, has increased [26,29]. Chronic wounds are also associated with decreased quality of life, high treatment costs, and morbidity [7,8]. The basic tenets of chronic wound care include debridement and proper dressing [9,10], but effective

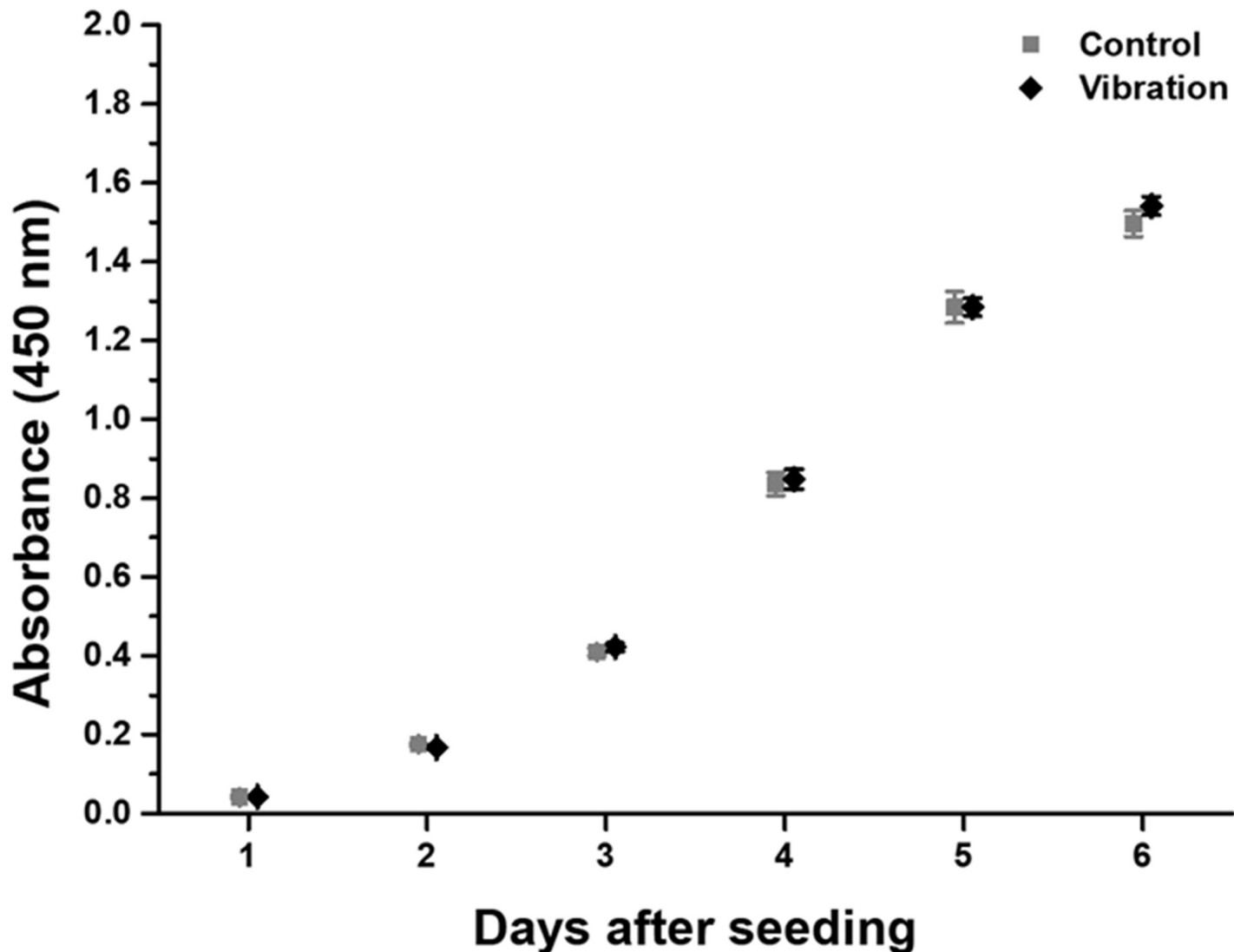

**Fig 4. Cell proliferative activity as assessed by the CCK-8 assay.** Every 24 h until the last absorbance measurement, absorbances at 450 nm were measured. The HaCaT cells were initially exposed to mechanical vibration for 2 h ($n$ = 6).

treatments have remained elusive. Although treatments are often carried out by applying one or several mechanical energy-based therapies to accelerate the wound healing process [11–16], there is a lack of understanding of the cytophysiological mechanisms in such mechanical energy-based therapies. In the current study, we hypothesized that a mechanical load affects mechano-transduction signaling pathways, leading to beneficial effects on the wound healing process. To test our hypothesis, we developed a vibrational cell culture model, and HaCaT cells were exposed to vibratory stimulation to determine the vibrational effects on cellular responses. In this study, we observed that the applied mechanical vibration induced changes in cytoskeleton, ERKs activation, and gene expression levels of some growth factors and growth factor receptors.

To investigate the wound healing effects of mechanical stimulation, we assessed cell viability and proliferation and performed *in vitro* scratch assays following exposure to mechanical vibration. Our results showed that the applied mechanical vibration did not induce cell death.

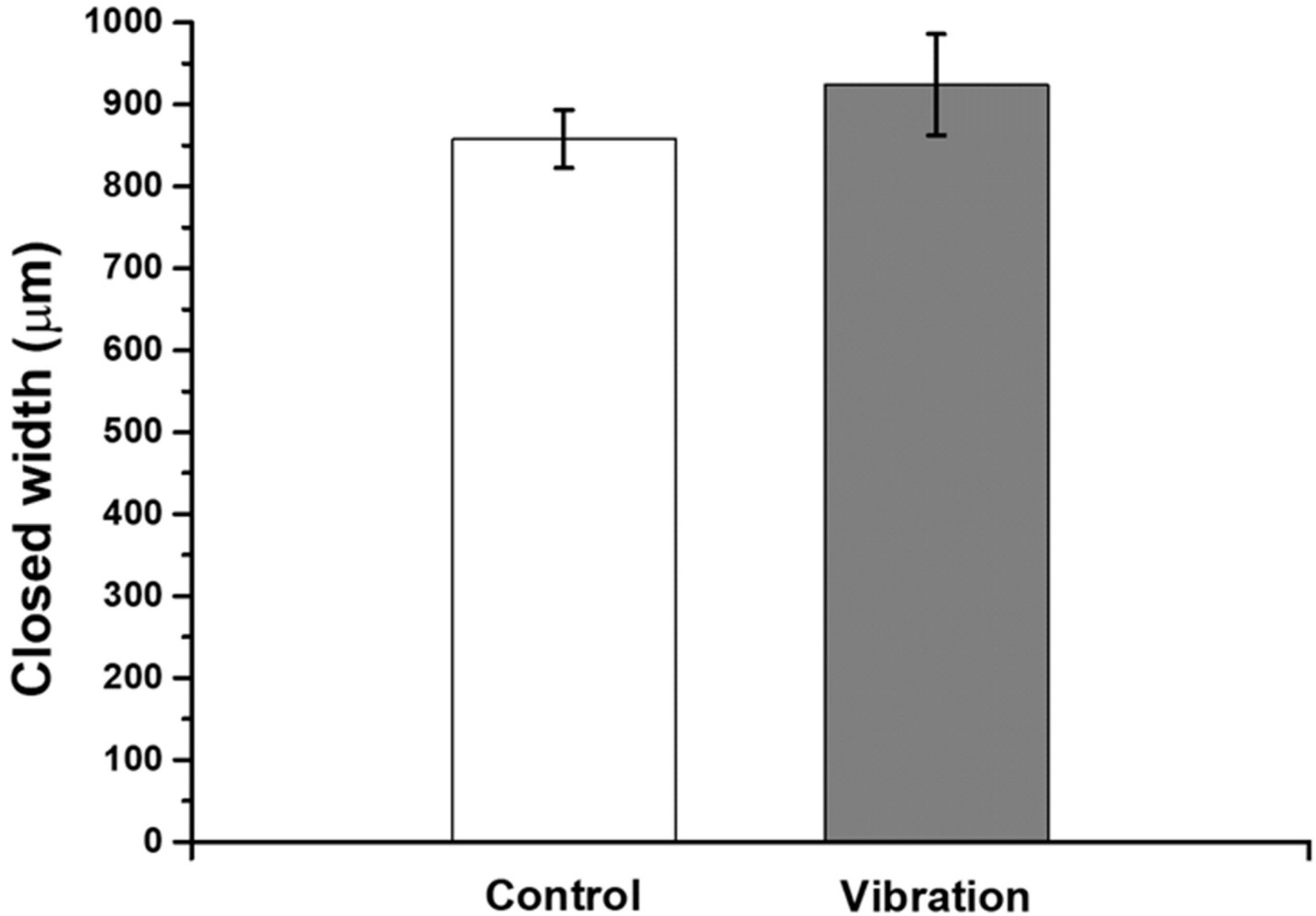

**Fig 5. Wound healing cell migration assay.** The HaCaT cell monolayers were scratched with a pipette tip and exposed to vibrational stimulation for 2 h per day. The closed width was calculated by subtracting the average length of vertical lines connecting the edges of the cell-free area measured at 96 hours after scratching from that measured immediately after scratching ($n$ = 4).

In addition, we observed that the applied vibration had no beneficial effect on cell proliferation or migration, both of which are important during the wound healing process. Some previous studies have reported on vibrational effects on cell proliferation and migration *in vitro* and on wound healing effects in animal tests. Lau *et al.* reported that a 60 Hz sinusoidal vibration at 0.3 *g* did not affect cell proliferation of rat mesenchymal stromal cells [30]. Kim *et al.* reported that cell proliferation of vocal fold tissue-derived cells was only slightly changed following exposure to 205 Hz vibrations [31,32]. In contrast, Zhang *et al.* showed that 50 Hz vibrations at accelerations of 0.3 *g* to 0.9 *g* decreased cell proliferation of human periodontal ligament stem cells [33]. Although these differences in reported results may be associated with variations in experimental conditions, such as cell types, culture media, culture period, and applied vibration levels, most of the *in vitro* studies have indicated that mechanical vibration does not promote cell proliferation. On the other hand, some animal studies have shown that low-intensity whole-body vibration can accelerate wound healing in diabetic mice and rats. Weinheimer-Haus *et al.* reported that a low-intensity vibration (0.4 *g* at 45 Hz) improved angiogenesis and wound healing in diabetic mice [15]. Yu *et al.* showed similar results in which an LMHFV

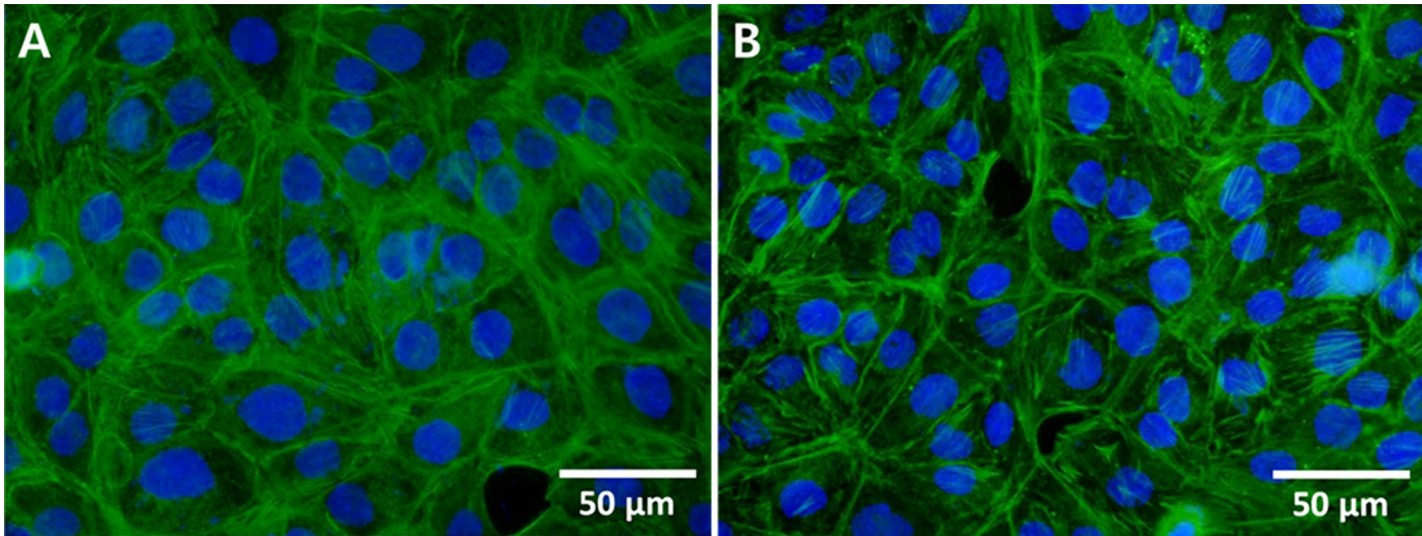

**Fig 6. Immunofluorescence images of F-actin in HaCaT cells.** F-actin was stained with AlexaFluor 488 conjugated phalloidin (green) and nuclei were stained with DAPI (blue). (A) Control group; (B) vibration-exposed group.

accelerated foot wound healing in diabetic rats by enhancing blood circulation and the glucose uptake rate [16]. These contradictory results between *in vitro* and animal studies may because the actual wound healing process does not only depend on the cell proliferation and migratory abilities of keratinocytes but also on their microenvironment components such as various growth factors and cytokines secreted from fibroblasts, platelets, macrophages, and keratinocytes, as well as nutrients provided via blood vessels [4,34].

Although we could not demonstrate vibrational effects on cell proliferation and migration, the aforementioned animal studies have shown there are wound healing effects of mechanical vibration. Thus, we studied cellular-level responses to vibration in order to investigate the potential wound healing benefits of mechanical vibration. Our results showed that the applied mechanical vibration induced reorganization of cytoskeleton structures. Filamentous actin structures were more tightly condensed, and they accumulated at the peripheral side near the cell membrane following exposure to vibration. Further, the western blot results indicated that the applied mechanical vibration activated ERK1/2 in the HaCaT cells. Actin filaments have important roles in sensing mechanical stimuli [35,36], and it has been suggested that actin cytoskeletal remodeling can mediate mechanical stress-induced physiological responses including gene expression and cell proliferation and differentiation [37]. Laboureau *et al.* reported that mechanical stresses, such as cell stretching, affect the small G proteins rac-1 and rhoA, which control organization of the cytoskeleton, leading to activation of ERKs, one of the well-known MAPKs [38]. Correa-Meyer *et al.* showed that mechanical cyclic stretching activates ERKs via G proteins and EGFR [39]. The activated ERKs translocate to the nucleus and activate various transcription factors, thereby changing gene expression and promoting growth, differentiation, or mitosis [22].

Subsequently, because the wound healing process is regulated by a complex signaling network of numerous growth factors, cytokines, and chemokines [40], we investigated the changes in gene expression levels of several growth factors and their receptors following exposure to mechanical vibration. FGF-2, also known as basic FGF, regulates ECM synthesis and deposition and enhances the motility of keratinocytes and fibroblasts [41,42]. PDGF induces mitogenicity and chemotaxis of fibroblasts, neutrophils, macrophages, and smooth muscle

**A**

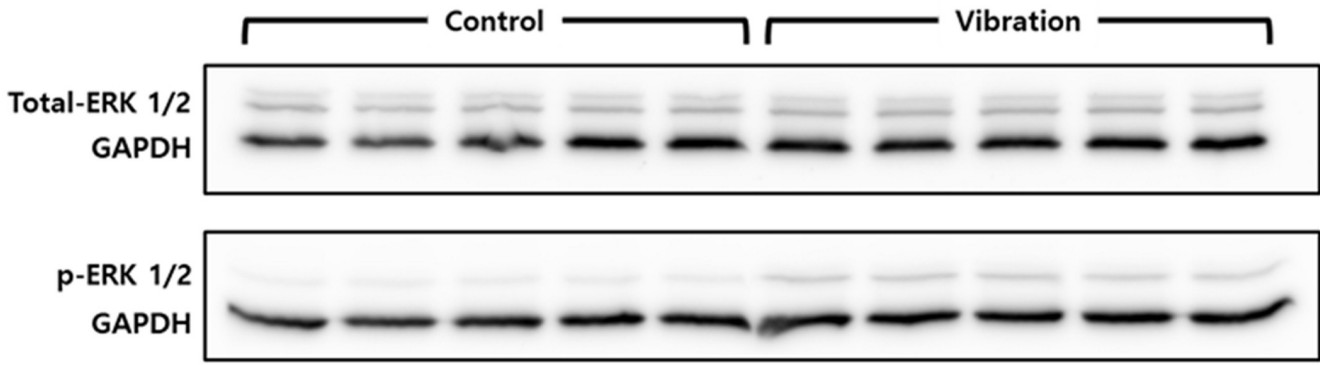

**B**

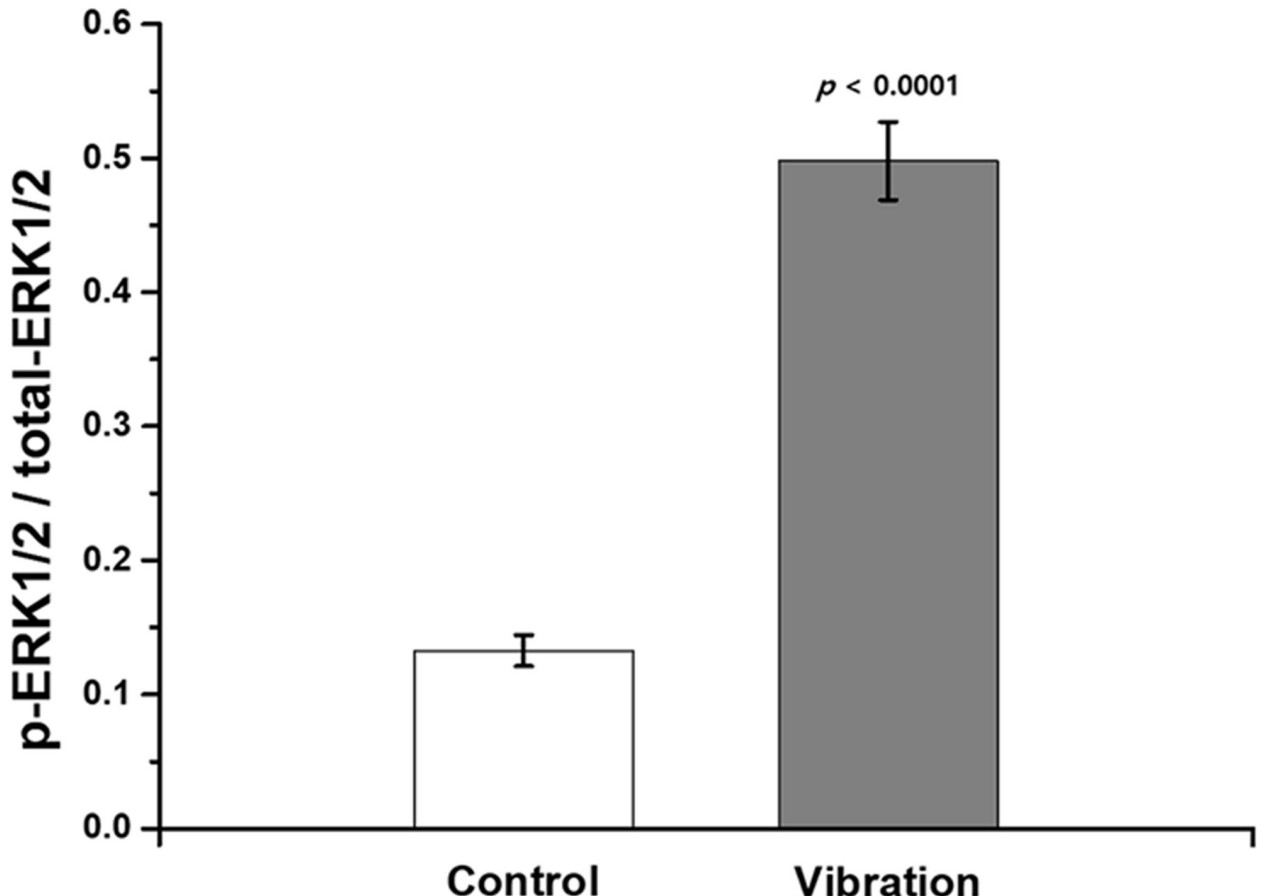

**Fig 7. Western blot analysis of ERK1/2 activation in HaCaT cells following exposure to mechanical vibration.** (A) Cropped images of western blot results on total-ERK1/2, p-ERK1/2, and GAPDH. Full length blots are presented in S1 Fig. (B) Relative levels of ERK1/2 activations obtained by performing densitometry analysis ($n = 5$).

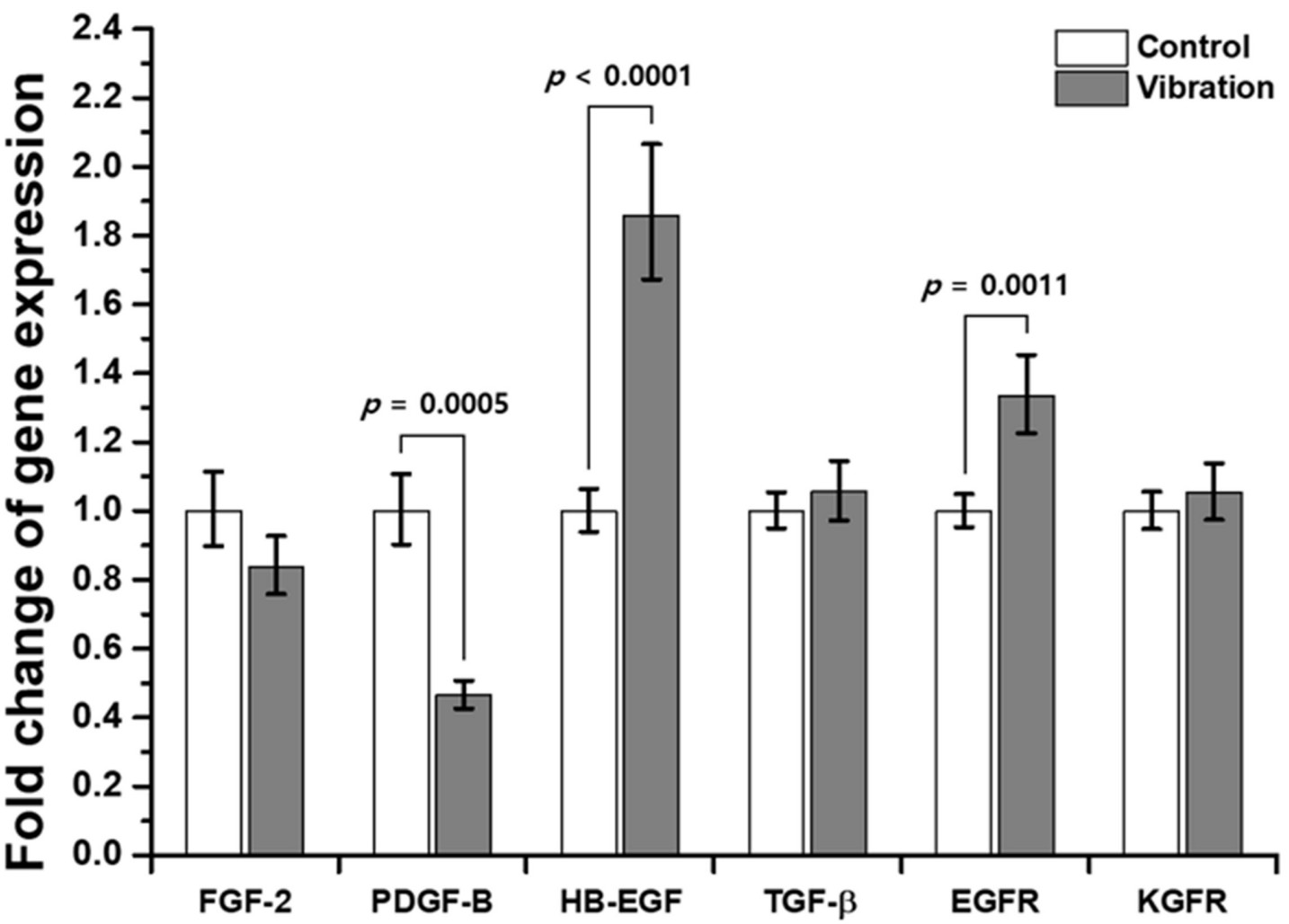

**Fig 8. Fold changes in gene expression levels of growth factors and their receptors in HaCaT cells following exposure to mechanical vibration.** Fold change was analyzed by using the $2^{(-\Delta\Delta CT)}$ method, and GAPDH was used as the housekeeping gene ($n = 4$).

cells to wound sites, and stimulates macrophages to synthesize and secrete growth factors [43,44]. Acting as a ligand to the EGFR, HB-EGF promotes the proliferation and migration of keratinocytes through autocrine signaling [45]. TGF-β is also important during the wound healing process and regulates cell proliferation, cell differentiation, and other functions [46]. Among the growth factor receptors, EGFR and KGFR have crucial roles in cell proliferation and differentiation, with both activating several signal transduction pathways including that of ERKs [47]. In the current study, we observed that the applied mechanical vibration increased the gene expression levels of HB-EGF and EGFR by 1.86-fold and 1.33-fold, respectively. Although the assessed targets were not identical to those in the current study, some other studies have also reported that LMHFV can increase the expression of growth factors, including insulin-like growth factor 1 and vascular endothelial growth factor [15,48]. Considering our results and those in previous studies (Table 2), we conclude that mechanical vibration triggers mechano-transduction signaling pathway responses that include cytoskeletal reorganization, ERKs activation, and regulation of gene expressions.

**Table 2. Comparison of the findings and limitations of studies related to the wound healing effects of vibration.**

| Study type | Vibratory condition | Findings | Limitations | Ref. |
|---|---|---|---|---|
| *in vivo* | 45 Hz (0.4 *g*) | • Increased angiogenesis and wound healing<br>• Reduced neutrophil accumulation and increased macrophage accumulation<br>• Increased IGF-1, VEGF, MCP-1 expressions | • Did not describe the wound healing mechanism at the cellular level | 15 |
| *in vivo* | 35 Hz (0.3 *g*) | • Increased blood microcirculation<br>• Reduced blood glucose level<br>• Increased GLUT4 expression | • Did not describe the wound healing mechanism at the cellular level | 16 |
| *in vitro* | 60 Hz (0.3 *g*) | • Did not affect cell proliferation | • Focused on osteogenic differentiation<br>• Used rat mesenchymal stromal cells | 30 |
| *in vitro* | 205 Hz (4 *g*) | • Increased metabolic activity (but only slightly) | • Focused on characterization of vocal fold-derived cells<br>• Did not use low-magnitude high-frequency vibratory condition<br>• Used vocal fold-derived cells | 31, 32 |
| *in vitro* | 50 Hz (0.05 *g* to 0.9 *g*) | • Decreased proliferation | • Focused on osteogenic differentiation<br>• Used human periodontal ligament stem cells | 33 |
| *in vitro* | 45 Hz (0.8 *g*) | • Induced cytoskeletal rearrangement<br>• Increased ERK1/2 activation<br>• Increased HB-EGF, EGFR expressions<br>• Increased cell migration (but the increase was not statistically significant) | • Did not demonstrate significant beneficial effects on cell proliferation and migration | This study |

IGF-1, insulin-like growth factor 1; VEGF, vascular endothelial growth factor; MCP-1, monocyte chemoattractant protein 1; GLUT4, glucose transporter type 4; ERK1/2, extracellular signal-regulated kinases 1 and 2; HB-EGF, heparin-binding epidermal growth factor-like growth factor; EGFR, epidermal growth factor receptor

In the current study, we developed a vibrational cell culture model and assessed the wound healing potential of mechanical vibration. Although our results did not demonstrate beneficial vibrational effects on cell proliferation and migration, we did observe that the applied mechanical vibration induced cytoskeletal rearrangement, activated ERKs, and increased the gene expressions of HB-EGF and EGFR. Taken together, the results of the current and previous studies suggest that mechanical vibration acts as a signal to the cytoskeleton, leading to activation of ERKs and regulation of gene expression, thereby promoting cell proliferation and migration. Our results indicate that mechanical vibration may possess wound healing potential; simultaneously, our results support those in previous studies on the wound healing effects of mechanical energy-based therapy.

## Supporting information

**S1 Fig. Full length images of western blots presented in the manuscript.** (A) Full length image of total-ERK1/2 and GAPDH; (B), (C) inverted and overexposed version of (A) to confirm the borderline of the polyvinylidene difluoride (PVDF) membrane; (D) full length image of p-ERK1/2 and GAPDH; (E), (F) inverted and overexposed version of (D) to confirm the borderline of the PVDF membrane; (G) western blot image of total-ERK1/2 and GAPDH with size markers. Red lines show the cropping locations.
(TIF)

## Author Contributions

**Conceptualization:** Dongjoo Kim, Soonjo Kwon.

**Data curation:** Dongjoo Kim, Soonjo Kwon.

**Formal analysis:** Dongjoo Kim, Soonjo Kwon.

**Funding acquisition:** Soonjo Kwon.

**Investigation:** Dongjoo Kim, Soonjo Kwon.

**Methodology:** Dongjoo Kim, Soonjo Kwon.

**Project administration:** Dongjoo Kim, Soonjo Kwon.

**Resources:** Dongjoo Kim, Soonjo Kwon.

**Supervision:** Soonjo Kwon.

**Validation:** Dongjoo Kim, Soonjo Kwon.

**Visualization:** Dongjoo Kim, Soonjo Kwon.

**Writing – original draft:** Dongjoo Kim.

**Writing – review & editing:** Dongjoo Kim, Soonjo Kwon.

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
