## [Decision Letter · Decision Letter 0]

29 Jan 2020

PONE-D-19-35448

Vibrational stress affects extracellular signal-regulated kinase activation and cytoskeleton structure in human keratinocytes

PLOS ONE

Dear Prof. Kwon,

Thank you for submitting your manuscript to PLOS ONE. After careful consideration, we feel that it has merit but does not fully meet PLOS ONE’s publication criteria as it currently stands. Therefore, we invite you to submit a revised version of the manuscript that addresses the points raised during the review process.

We would appreciate receiving your revised manuscript by Mar 14 2020 11:59PM. To enhance the reproducibility of your results, we recommend that if applicable you deposit your laboratory protocols in protocols.io, where a protocol can be assigned its own identifier (DOI) such that it can be cited independently in the future. For instructions see: http://journals.plos.org/plosone/s/submission-guidelines#loc-laboratory-protocols

We look forward to receiving your revised manuscript.

Kind regards,

Vivek K. Bajpai, PhD

Academic Editor

PLOS ONE

Additional Editor Comments:

Dear Authors,

We have now received required numbers of reviews on your manuscript, and you can see that reviewers have asked to revise the article. Therefore, I would like to invite you to revise your manuscript based on the comments of the reviewers and submit your revised manuscript as soon as possible along with a separate response letter to the comments of the reviewers.

I would like to ask you to recheck your manuscript for scientific English editing while submitting the revised manuscript.

Looking forward to receiving your revised manuscript.

Thank you very much for choosing PlosOne as the publication platform to your scientific work.

Kind regards,

Vivek K. Bajpai

Reviewers' comments:

Reviewer's Responses to Questions

**Comments to the Author**

1. Is the manuscript technically sound, and do the data support the conclusions?

Reviewer #1: Yes

Reviewer #2: Yes

2. Has the statistical analysis been performed appropriately and rigorously? 

Reviewer #1: Yes

Reviewer #2: Yes

3. Have the authors made all data underlying the findings in their manuscript fully available?

Reviewer #1: Yes

Reviewer #2: Yes

4. Is the manuscript presented in an intelligible fashion and written in standard English?

Reviewer #1: Yes

Reviewer #2: No

5. Review Comments to the Author

Reviewer #1: Dear editor:

Recommendation: Minor revisions needed as noted.

I have thoroughly reviewed the manuscript entitled "Vibrational stress affects extracellular signal-regulated kinase activation and cytoskeleton structure in human keratinocytes". This manuscript describes the effects of mechanical vibration stress on human keratinocytes using an in vitro vibrational cell culture model. The manuscript is well written, the experiments are clearly presented the reference list is adequate. This research can be promising mechanical energy-based treatment for enhancing wound healing efficiency, as a result I strongly recommend the publication of the manuscript as it is. Additionally, some suggestions were listed as follows for the authors to improving the work.

Minor comments:

# 1. This manuscript investigated the effects of mechanical vibration using the in vitro model. Experimental results of this work are interesting, although the innovative points of this paper did not demonstrated clearly in the introduction part, experimental section, and results and discussions is not clearly presented. Thus, I suggest that the authors summarize the specific characteristics and differences of proposed method in the form of a Table by comparing conventional methods.

# 2. The authors should improve their paper-writing with correct grammar and scientific scope.

Reviewer #2: Manuscript designed with scientific and technical merits However few minor modifications are required.

1. In abstract kindly mention the specific immuno-stains and protein markers to perform the molecular pathway.

2. Cross check for mentioning full name followed by abbreviations.

3. Conclusion should cover pros and cons of the outcome of research.

4. Scientific expressions in terms of English should be edited by any science subject experts at professional level.

5. Check spell errors throughout the manuscript.

6. PLOS authors have the option to publish the peer review history of their article (what does this mean?). If published, this will include your full peer review and any attached files.

Reviewer #1: No

Reviewer #2: No

---

## [Author Response · Author response to Decision Letter 0]

22 Feb 2020

February 22, 2020

We thank the reviewers for the insightful and constructive suggestions. We are addressing them below. The revised manuscript has been much improved. The reviewers’ comments are in italic, while responses are in plain text.

Reviewer #1’s comments:

I have thoroughly reviewed the manuscript entitled "Vibrational stress affects extracellular signal-regulated kinase activation and cytoskeleton structure in human keratinocytes". This manuscript describes the effects of mechanical vibration stress on human keratinocytes using an in vitro vibrational cell culture model. The manuscript is well written, the experiments are clearly presented the reference list is adequate. This research can be promising mechanical energy-based treatment for enhancing wound healing efficiency, as a result I strongly recommend the publication of the manuscript as it is. Additionally, some suggestions were listed as follows for the authors to improving the work.

1) This manuscript investigated the effects of mechanical vibration using the in vitro model. Experimental results of this work are interesting, although the innovative points of this paper did not demonstrated clearly in the introduction part, experimental section, and results and discussions is not clearly presented. Thus, I suggest that the authors summarize the specific characteristics and differences of proposed method in the form of a Table by comparing conventional methods.

Response: We thank the reviewer for the comment. As the reviewer suggested, we summarized the characteristics and differences of the current study compared to previous ones in the form of a table (Please see Table 2 in Discussion).

We added it in Discussion section of current manuscript.

2) The authors should improve their paper-writing with correct grammar and scientific scope.

Response: As the reviewer suggested, we corrected grammar and scientific scope throughout the manuscript.

Reviewer #2’s comments:

Manuscript designed with scientific and technical merits However few minor modifications are required.

1) In abstract kindly mention the specific immuno-stains and protein markers to perform the molecular pathway.

Response: We added the specific immunostaining and western blot markers in that sentences for the relevant information in Abstract as the reviewer suggested.

“ ~ The results of immunostaining (F-actin), western blot (ERK1/2), and RT-qPCR (FGF-2, PDGF-B, HB-EGF, TGF-β1, EGFR, and KGFR) analyses demonstrated that the applied vibration resulted in rearrangement of the cytoskeleton, leading to activation of ERK1/2, one of the MAPK signaling pathways, and upregulation of the gene expression levels of HB-EGF and EGFR.~”

2) Cross check for mentioning full name followed by abbreviations.

Response: We thank the reviewer for the comment. As the reviewer suggested, we checked abbreviations and their full names, and revised some errors.

3) Conclusion should cover pros and cons of the outcome of research.

Response: We agreed with reviewer’s comment. As you suggested, we added the related contents in the Discussion section in the form of a table.

4) Scientific expressions in terms of English should be edited by any science subject experts at professional level.

Response: As you suggested, we revised and improved scientific expressions throughout the manuscript.

5) Check spell errors throughout the manuscript.

Response: As you suggested, we checked spell errors and revised the manuscript.

We also corrected the reference #30.

---

## [Editor Report · Decision Letter 1]

18 Mar 2020

Vibrational stress affects extracellular signal-regulated kinase activation and cytoskeleton structure in human keratinocytes

PONE-D-19-35448R1

Dear Dr. Kwon,

We are pleased to inform you that your manuscript has been judged scientifically suitable for publication and will be formally accepted for publication once it complies with all outstanding technical requirements.

With kind regards,

Vivek K. Bajpai, PhD

Academic Editor

PLOS ONE

Additional Editor Comments (optional):

The manuscript has been improved well, thus I recommend it for publication.

Vivek K. Bajpai,

AE, PlosOne
---

## [Editor Report · Acceptance letter]

24 Mar 2020

PONE-D-19-35448R1 

Vibrational stress affects extracellular signal-regulated kinases activation and cytoskeleton structure in human keratinocytes 

Dear Dr. Kwon:

I am pleased to inform you that your manuscript has been deemed suitable for publication in PLOS ONE. Congratulations! Your manuscript is now with our production department. 

With kind regards,

on behalf of

Dr. Vivek K. Bajpai 

Academic Editor

PLOS ONE